# Treatment of Anemia Associated with Chronic Kidney Disease: Plea for Considering Physiological Erythropoiesis

**DOI:** 10.3390/ijms25137322

**Published:** 2024-07-03

**Authors:** Takahiro Kuragano

**Affiliations:** Division of Kidney and Dialysis, Hyogo Medical University, Hyogo 663-8501, Japan; kuragano@hyo-med.ac.jp; Tel.: +81-798-45-6521; Fax: +81-798-45-6880

**Keywords:** anemia with chronic kidney disease, hyporesponsiveness to ESAs, defective iron utilization for erythropoiesis, physiological erythropoiesis

## Abstract

Traditionally, the treatment of anemia associated with chronic kidney disease (CKD) involves prescribing erythropoiesis-stimulating agents (ESAs) or iron preparations. The effectiveness and safety of ESAs and iron have been established. However, several clinical issues, such as hyporesponsiveness to ESAs or defective iron utilization for erythropoiesis, have been demonstrated. Recently, a new class of therapeutics for renal anemia known as hypoxia-inducible factor (HIF)/proline hydroxylase (PH) inhibitors has been developed. Several studies have reported that HIF-PH inhibitors have unique characteristics compared with those of ESAs. In particular, the use of HIF-PH inhibitors may maintain target Hb concentration in patients treated with a high dose of ESAs without increasing the dose. Furthermore, several recent studies have demonstrated that patients with CKD with defective iron utilization for erythropoiesis had a high risk of cardiovascular events or premature death. HIF-PH inhibitors increase iron transport and absorption from the gastrointestinal tract; thus, they may ameliorate defective iron utilization for erythropoiesis in patients with CKD. Conversely, several clinical problems, such as aggravation of thrombotic and embolic complications, diabetic retinal disease, and cancer, have been noted at the time of HIF-PH inhibitor administration. Recently, several pooled analyses of phase III trials have reported the non-inferiority of HIF-PH inhibitors regarding these clinical concerns compared with ESAs. The advantages and issues of anemia treatment by ESAs, iron preparations, and HIF-PH inhibitors must be fully understood. Moreover, patients with anemia and CKD should be treated by providing a physiological erythropoiesis environment that is similar to that of healthy individuals.

## 1. Introduction

Erythropoietin (EPO) is a glycoprotein with a molecular weight of approximately 34,000 to 46,000 that stimulates the differentiation of erythroid stem cells and promotes red blood cell production. Erythropoiesis-stimulating agents (ESAs) were approved as recombinant human erythropoietin (rHuEPO) agents for anemia treatment in the late 1980s. Iron is essential for the transport of oxygen by hemoglobin (Hb) and myoglobin and contributes to the activity of several enzymes that are involved in the mitochondrial respiratory chain and DNA synthesis in mammalian species. Therefore, ESAs and iron supplements have traditionally been used for the treatment of anemia associated with chronic kidney disease (CKD). The effectiveness and safety of ESAs and iron have been previously established. However, in recent years, several large cohort studies have demonstrated that patients who are hyporesponsive to ESAs or who have defective iron utilization for erythropoiesis have a greater risk of adverse events or premature death. Furthermore, a high dose of ESA or iron preparation is associated with adverse events or premature death in patients with CKD. Recently, hypoxia-inducible factor (HIF)/proline hydroxylase (PH) inhibitors have become available for clinical use. HIF-PH inhibitors promote the dimerization and stabilization of HIF-α and HIF-β in the nucleus by inhibiting proline hydroxylase (PH), which degrades HIF-α and stimulates endogenous erythropoietin (EPO) production by inducing EPO gene expression, thereby promoting erythropoiesis. Furthermore, inducing the expression of the iron-regulating factor divalent metal transporter (DMT) 1, duodenal cytochrome B (DCytB), transferrin, transferrin receptor (TfR), and ceruloplasmin increases iron transport and absorption from the gastrointestinal tract, thereby creating an environment in which stored iron in the whole body can be effectively utilized for erythropoiesis [1]. Therefore, in phase III trials, HIF-PH inhibitors have demonstrated characteristics different from those of conventional drugs for anemia associated with CKD and have been found to be effective and safe.

In this review, we discuss the difference in erythropoiesis patterns between patients receiving ESAs and those receiving HIF-PH inhibitors and the impact of erythropoiesis patterns on adverse events and survival in patients with CKD.

## 2. Effectiveness and Issues of Anemia Treatments with ESAs

### 2.1. Effectiveness of ESAs

Erythropoietin (EPO) is a glycoprotein with a molecular weight of approximately 34,000 to 46,000 that stimulates the differentiation of erythroid stem cells and promotes red blood cell production. EPO is regulated by the oxygen partial pressure of arterial blood and is produced by renal erythropoietin-producing (REP) cells in fibroblasts of the interstitium in the renal cortex. Thus, relative EPO deficiency is thought to be the main cause of renal anemia. ESAs were approved as rHuEPO agents for anemia treatment in the late 1980s. Furthermore, darbepoetin alfa and epoetin beta pegol were also approved as long-acting ESAs. Approximately 30 years have passed since the introduction of ESAs for the treatment of anemia in patients with CKD, and the efficacy and safety of these treatments have been well established.

### 2.2. Issues of Hyporesponsiveness to ESAs

Several studies have reported that patients with CKD with hyporesponsiveness to ESAs are at high risk of adverse events such as cardiovascular complications and premature death [2]. However, few articles and reviews have clearly defined the criteria for hyporesponsiveness to ESAs. Malnutrition, deficiency of trace elements (such as iron, vitamin B12, folic acid, and zinc), blood disorders, malignant tumors, uremia (inadequate dialysis), chronic inflammation, hypothyroidism, hyperparathyroidism, and various drugs (such as renin–angiotensin system inhibitors) are known to affect ESA responsiveness. Recently, a global randomized cardiovascular outcome study [3] comparing HIF-PH inhibitors with ESAs performed with 2694 patients on hemodialysis who were iron replete at baseline reported that 12% of patients were defined as ESA hyporesponsive. According to a multivariable regression analysis, a lower body mass index, lower TSAT, younger age, and lower serum albumin levels, as well as a higher dose of intravenous iron, were identified as being strong predictors of ESA hyporesponsiveness. Furthermore, in a cross-sectional study [4] of 4460 patients who had nondialysis-dependent CKD, older age, female sex, lower body mass index, lower cholesterol, lower glomerular filtration rate, and higher intact parathyroid hormone levels were identified as being independent predictors of ESA hyporesponsiveness. Moreover, it has been reported that accumulation of uremic toxins such as indoxyl sulfate in blood impairs erythropoietin synthesis, compromising the growth and differentiation of red blood cells in bone marrow, leading to a subsequent impairment of erythropoiesis. Thus, the physiopathology of anemia in CKD focusing on a potential mechanism of EPO on iron regulation in a uremic condition might also be important [5].

Therefore, the management of anemia caused by an ESA hyporesponse is multifaceted. The control of inflammatory conditions, improvements in nutritional conditions, maintenance of adequate dialysis doses, and maintenance of optimal iron status are needed.

### 2.3. Issues Associated with the High-Dose Administration of ESAs

In an observational study of patients on hemodialysis (TRAP) over 2 years [6], we reported that the risks of infectious disease and hospitalization, even after correction for Hb, ferritin, albumin, CRP levels, age, sex, and etiology, were greater in the high-dose ESA (≥4500 IU/week) group than in the low-dose ESA group (<4500 IU/week). Furthermore, recent epidemiological studies of 6792 patients with CKD (not on dialysis) and 2844 patients on hemodialysis have reported that high-dose ESA administration is associated with a risk of major adverse cardiovascular events (MACEs: stroke, nonfatal myocardial infarction, nonfatal heart failure, unstable angina, and total mortality), independent of hyporesponsiveness to ESAs [7]. A high dose of ESAs may induce strong fluctuations in serum levels of erythropoietin (EPO) that exceed the physiological range [8]. Furthermore, a high dose of ESAs may increase platelet counts by inducing megakaryocyte growth and differentiation and stimulate platelet activity by inducing von Willebrand factor release, as well as E-selectin and P-selectin expression [9]; moreover, these effects induced by a high dose of ESAs may cause thromboembolic complications. Therefore, when considering physical erythropoiesis, treatment with a high dose of ESAs may be better to avoid in patients with CKD.

## 3. Effectiveness and Issues of Anemia Treatments with Iron Preparation

### 3.1. Effectiveness of Iron Preparations

Most of the protein contained in red blood cells is in the form of Hb. Iron is essential for the transport of oxygen by Hb and myoglobin and contributes to the activity of several enzymes involved in the mitochondrial respiratory chain and DNA synthesis in mammalian species. Therefore, sufficient amounts of iron are needed for erythropoiesis. Appropriate iron administration to patients with absolute iron deficiency induces effective erythropoiesis and corrects anemia in patients with CKD.

It is well known that absolute iron deficiency may cause several clinical problems, including not only anemia but also low physical performance, dysregulation of thermoregulation, neurocognitive dysfunction, restless leg syndrome, immune dysfunction, koilonychias, and Plummer–Vinson syndrome. Furthermore, in heart failure patients with iron deficiency, intravenous administration of 200 mg of ferric carboxymaltose every 4 weeks for 24 weeks reportedly improved symptoms related to heart failure [10]. Furthermore, a multicenter randomized controlled trial of 2141 patients on hemodialysis [11] demonstrated that, compared with those in a low-dose intravenous iron supplementation group (0–400 mg/month), those in a high-dose intravenous iron supplementation group (400 mg/month) exhibited a lower risk of death and nonfatal myocardial infarction. This study also demonstrated that patients who enrolled in the proactive high-dose arm of this non-inferiority trial (per-protocol analysis) received an average monthly dose of 264 mg, whereas those in the low-dose reactive arm received 145 mg monthly, thus confirming prospectively the lack of toxicity in patients on hemodialysis with an intravenous iron dosage below 300 mg/month, which had been shown by DOPPS to be associated with mortality in this setting.

### 3.2. Issues of Defective Iron Utilization for Erythropoiesis

Defective iron utilization is a condition in which iron is not effectively utilized for erythropoiesis even if enough iron is stored in the whole body. Hepcidin is produced by hepatocytes, circulates in the plasma, and is considered to play a central role in regulating iron availability. Hepcidin binds to ferroportin, which is a cellular iron export channel protein, thus causing it to be internalized and degraded in lysosomes, which then prevents the efflux of iron from iron-exporting tissues into the plasma. It is suspected that an excess of hepcidin leads to defective iron utilization for erythropoiesis in patients with CKD. It has been suspected that impaired renal function, inflammation, uremia, and excessive iron administration may cause high serum levels of hepcidin in patients with CKD. Previously, we reported that compared with those of healthy volunteers, serum levels of hepcidin in patients on maintenance hemodialysis were significantly greater. According to a multivariate analysis, only transferrin and ferritin were found to be significant predictors of hepcidin in both patients on hemodialysis and healthy volunteers. However, in patients on hemodialysis with inflammation (sensitive C-reactive protein > 0.3 mg/dL), ferritin and IL-6 levels were found to be significant predictors of serum hepcidin. Thus, we presumed that in the absence of apparent inflammation, the serum hepcidin level could be exclusively associated with iron storage (serum ferritin levels) in patients on hemodialysis and was independent of levels of inflammatory cytokines. However, in the presence of microinflammation, IL-6 may also affect hepcidin expression [12]. Hepcidin overexpression causes iron, which is used for erythropoiesis, to deposit in various tissues or organs. Indeed, a multiple regression analysis demonstrated that serum hepcidin levels in patients on hemodialysis are independently associated with brachial-ankle (ba) pulse wave velocity (PWV), which indicates vascular stiffness in these patients, as well as age, serum levels of total cholesterol, and tumor necrosis factor-α [13]. The exact mechanism by which hepcidin promotes arterial stiffness has not been clarified. However, several mechanisms have been proposed for the relationship between iron accumulation and arterial alteration, including the generation of oxidized LDL, endothelial cell dysfunction, and arterial smooth muscle proliferation. Recent basic and clinical research has demonstrated that defective iron utilization, as well as a relative deficiency in EPO, may be associated with the development and progression of anemia in patients with CKD. We previously reported that although the mRNA levels of EPO in the kidneys were significantly greater, compared with control mice, Hb was lower in adenine-induced CKD mice. Additionally, in adenine-induced CKD mice, serum levels of iron and TSAT are significantly decreased, and serum ferritin and hepcidin levels are significantly increased [14]. According to a cross-sectional study of 80 patients with CKD in the early stage, despite a significant increase in serum ferritin concentration, Hb levels were significantly decreased as a result of the decline in kidney function [15]. From these results, we presumed that defective iron utilization for erythropoiesis may be associated with the onset or progression of anemia in patients with CKD, as well as with relative EPO deficiency. Furthermore, several recent studies have reported on the relationship between defective iron utilization for erythropoiesis and adverse events in patients with CKD. In a 3-year observational study of 805 patients on maintenance hemodialysis, we reported that patients with defective iron utilization for erythropoiesis (high ferritin [≥100 ng/mL] and low TSAT [<20%]) had significantly greater risks of cerebrovascular and cardiovascular disease and death than patients with low ferritin (<100 ng/mL) and high TSAT (≥20%) [16]. Several studies have reported of a greater risk for all-cause and cardiovascular-specific mortality in patients with lower TSAT levels. A lower TSAT may reflect iron deficiency. However, in our study, patient sex, age, comorbidities (such as diabetes), hCRP levels, and β2-MG levels were identified as being significant predictors of TSAT levels in patients on hemodialysis, whereas the dosage of intravenous iron or iron storage (such as serum ferritin levels) was not identified as being a predictor of TSAT levels. Thus, we presumed that defective iron utilization for erythropoiesis has a stronger impact on adverse events in patients on hemodialysis than iron deficiency. Furthermore, in a 10-year retrospective observational study of 1173 patients on peritoneal dialysis, iron overload (high TSAT [>30%], high ferritin [>500 ng/mL]) and defective iron utilization for erythropoiesis (low TSAT [<20%] and high ferritin [>100 ng/mL]) predicted a high risk of all-cause mortality and cardiovascular mortality in patients on peritoneal dialysis [17]. Among 933,463 patients with CKD (not on dialysis), defective iron utilization for erythropoiesis (low TSAT [<20%] and high ferritin [100–500 ng/dL]) was associated with increased risks of mortality and cardiovascular hospitalization; however, absolute iron deficiency was associated only with a greater risk of hospitalization [18]. Even if iron supplementation in patients with defective iron utilization results in temporary amelioration of anemia, most of the administered iron is deposited in various tissues or organs, such as reticuloendothelial cells, which may lead to iron overload. Therefore, when considering long-term survival, iron administration to patients with CKD with defective iron utilization for erythropoiesis should be carefully performed.

### 3.3. Iron Overload and High-Dose Iron Administration

Sufficient amounts of iron are needed for erythropoiesis. However, free iron is of pivotal importance as a pro-oxidant cofactor associated with the production of hydroxyl radicals in cells, which damage systems that are vital for oxidation. In addition, free iron may be important for bacterial growth and could impact virulence to the host, which may be due in part to the physiology of iron sequestration in the face of infections. Therefore, iron needs to be tightly regulated under physiological conditions. The Dialysis Outcomes and Practice Patterns Study (DOPPs), which was performed on 32,435 patients on hemodialysis in 12 countries, demonstrated that the risks of all-cause mortality and cause-specific mortality (cardiovascular, infectious, and other) were significantly greater in patients treated with a high dose of intravenous iron (300–399 mg/month and 400 mg/month or more) than in patients treated with intravenous iron at 100–199 mg/month [19]. According to a 2-year observational study of 1086 Japanese patients on maintenance hemodialysis, the risk of cerebral cardiovascular disease, infection, and hospitalization was significantly greater in patients treated with high weekly doses of intravenous iron (>50 mg/week) than in patients not receiving iron [6]. There is a high incidence of iron overload detected by liver magnetic resonance imaging among maintenance patients on hemodialysis who are treated with intravenous iron preparations [20]. The results of these studies suggest that persistent administration of a higher dose of iron preparation should be avoided, as should a high dose of ESAs. The Dialysis Patients Response to IV Iron with Elevated Ferritin (DRIVE) study [21] reported of the ability of intravenous iron to improve Hb levels in patients on hemodialysis who had ferritin levels of 500–1200 ng/mL and TSAT levels under 25% who were suspected to have dysutilized iron for erythropoiesis. It is possible that iron administration to patients with dysutilization of iron for erythropoiesis may increase Hb levels, even if they have excess iron stores. On the other hand, we demonstrated that among patients on hemodialysis with hyporesponsiveness to ESAs, high ferritin levels (≥100 ng/mL) and the administration of a high dose of intravenous iron (≥50 mg/week) were associated with a greater risk of composite events (CCVD, infection, need for hospitalization, and death) than in other patients [22]. From the results of these studies, we presumed that although iron administration to patients with replete iron stores improves ESA responsiveness and maintains target Hb levels, iron administration to these patients may not necessarily reduce the long-term risk of adverse events or improve survival. In patients with hyporesponsiveness to ESAs who do not maintain the target Hb, increasing the dose of ESAs or iron is not unfavorable, and anemia treatment involving the consideration of iron utilization for erythropoiesis is needed.

## 4. Effectiveness of and Issues Associated with Anemia Treatments with HIF-PH Inhibitors

### 4.1. Effectiveness of the HIF-PH Inhibitor

Post hoc analysis of phase III trials of roxadustat (which is a HIF-PH inhibitor) showed that the dose of roxadustat that is required to maintain target hemoglobin (Hb) levels is not strongly affected by ESA responsiveness, high-sensitivity CRP, the Geriatric Nutritional Risk Index (GNRI), or iron-related factors [23]. According to these phase III trials, HIF-PH inhibitors can maintain Hb levels within the target range in dialysis patients who require high doses of ESAs without increasing their dose or iron administration. Thus, the prescription of HIF-PH inhibitors to these patients may prevent the iatrogenic risk of the toxic effects of increasing the dosage of iron preparations or ESAs. Furthermore, phase 3 trials have reported that, compared with conventional ESAs, most HIF-PH inhibitors significantly suppressed the serum levels of hepcidin, which induced defective iron utilization for erythropoiesis [24,25,26]. These results suggested that HIF-PH inhibitors may improve the defective iron utilization of patients with CKD and induce physiological erythropoiesis. Additional studies are needed to determine whether the administration of HIF-PH inhibitors to patients with defective iron utilization maintains stable target Hb levels and whether improvements in defective iron utilization lead to the suppression of adverse events or improve the prognosis of these patients.

### 4.2. Issues of Anemia Treatment with HIF-PH Inhibitors

The long-term safety of HIF-PH inhibitor administration must be confirmed. In particular, HIF-PH inhibitors may stimulate vascular endothelial growth factor, which is an angiogenic factor, and plasminogen activator inhibitor-1, which is a coagulation and fibrinolytic factor [27]. Therefore, careful prescription of HIF-PH inhibitors is required for patients with diabetic retinal disease or cancer. There is also concern about thrombotic and embolic complications that are associated with the administration of HIF-PH inhibitors. A pooled analysis of phase III trials conducted with multiple HIF-PH inhibitors has been published, and it compared the risk of MACEs and all-cause mortality in a group of patients treated for anemia with conventional ESAs or placebo and HIF-PH inhibitors. Although there were some discrepancies in the results of these studies, vadadustat and daprodustat were found to be non-inferior in terms of the risk of MACEs or survival compared with conventional ESAs in patients on hemodialysis. The INNO2VATE Clinical Trial [28], which was performed in 3923 patients on hemodialysis, showed that there was no significant difference in the risk of MACEs (death [any cause], nonfatal myocardial infarction, or nonfatal stroke), expanded MACEs (MACE+ hospitalization for heart failure or thromboembolic event, excluding vascular access failure), death from cardiovascular cause, or all-cause death between the patients treated with vadadustat and those treated with darbepoetin. On the other hand, the PRO2TECT Clinical Trials [29], which were conducted on patients with CKD who were not on dialysis, failed to demonstrate the non-inferiority of vadadustat to darbepoetin in terms of risk of MACEs. Furthermore, a sub-analysis of this study demonstrated non-inferiority in the risk of MACEs or expanded MACEs of vadadustat compared with darbepoetin studies conducted in the United States, but vadadustat’s non-inferiority was not demonstrated outside the United States. Moreover, the ASCEND-D [30] study performed in 2964 patients on hemodialysis indicated that there was no significant difference in the risk of MACEs or thromboembolic events; MACEs or hospitalization for heart failure; or all-cause death between patients treated with daprodustat and those treated with conventional ESAs. A pooled analysis based on four studies (HIMALAYAS, ROCKIES, SIERRAS, and PYRENEES), which were performed with 4726 patients on hemodialysis, reported that in stable patients on hemodialysis treated with roxadustat, HR point estimates for time to first MACE, MACE+, and all-cause mortality were slightly greater than those among patients treated with conventional ESAs [31,32]. The reason for the discrepancies in the risk of thrombotic and embolic complications among HIF-PH inhibitors is not clear. At present, we should prescribe HIF-PH inhibitors to patients with CKD after considering the risks of thrombotic and embolic complications, as well as the use of ESAs. Additionally, a pooled analysis of phase III trials of patients with CKD who were not on dialysis and patients on dialysis demonstrated no significant differences in the development or worsening of retinal hemorrhage or retinal thickness between the groups treated with roxadustat and those treated with darbepoetin alfa [33]. Furthermore, the risk of cancer is a concern when anemia is treated with HIF-PH inhibitors and ESAs. Recently, it was reported that in a phase III trial of daprodustat (which is a HIF-PH inhibitor) for patients on dialysis and patients with CKD who were not on dialysis, the use of daprodustat was not associated with an increased risk of cancer or cancer mortality relative to that associated with ESA use [34]. Conversely, a recent study demonstrated that treating several patients with CKD with roxadustat decreased thyroid-stimulating hormone (TSH) and free thyroxine (FT)4 levels [35]. Although the mechanism by which roxadustat reduces TSH and FT4 levels is unclear, additional studies on thyroid function may be needed for the treatment of anemia with a HIF-PH inhibitor. Although additional large-scale and long-term studies are needed, we should prescribe HIF-PH inhibitors after considering the balance between the benefits and risks of HIF-PH inhibitors.

## 5. Differences in the Erythropoiesis Environment between HIF-PH Inhibitors and ESAs

In phase III trials of vadadustat (which is a HIF-PH inhibitor) and darbepoetin alfa conducted in Japan among patients with CKD who were not on dialysis, serum ferritin, hepcidin, and TSAT levels were lower, and total iron binding capacity (TIBC) was greater in patients treated with vadadustat than in those treated with darbepoetin alfa. These changes in ferritin and TSAT levels seem to demonstrate vadadustat-induced iron deficiency via erythropoiesis. However, despite decreases in mean corpuscular volume (MCV) and mean corpuscular hemoglobin (MCH) in patients treated with darbepoetin alfa, mean corpuscular volume (MCV) and mean corpuscular hemoglobin (MCH) remained greater in patients treated with vadadustat [25]. These differences in erythrocyte size (MCV or MCH) and Hb levels between patients treated with vadadustat and those treated with darbepoetin alfa are consistent with improved iron delivery to erythroblasts, wherein increased intracellular iron enhances heme synthesis, which correspondingly increases Hb levels and erythrocyte size [36]. Furthermore, the authors of this study suggested that due to HIF increasing the transcription of the erythroid-specific *ALAS-2* gene [37] and iron-regulatory protein increasing the translation of ALAS-2 mRNAs, the administration of vadadustat increased the level of heme in erythroblasts. Increased erythroblast heme induces enhanced heme-regulated inhibitor-controlled synthesis of globins and other proteins [36], thus leading to the production of larger erythrocytes containing more Hb. Thus, these results suggest that the administration of vadadustat consumes stored iron; however, there is enough iron in red blood cells for this effect, and iron in the whole body is effectively used for erythropoiesis.

The same study also showed that red blood cell distribution width (RDW) was lower in patients treated with vadadustat than in those treated with darbepoetin alfa. An observational study of 1688 patients diagnosed with stage 3b-5 CKD reported that high RDWs or increased RDWs are associated with high mortality [38]. Moreover, a recent Japanese cohort study of 1320 patients who were not on dialysis for 4.7 years demonstrated that, compared to patients with high RDWs (≥13.6%), patients with low RDWs exhibited a significantly lower risk of end-stage kidney disease and all-cause death [39]. Aging, hyporesponsiveness to ESAs, and iron deficiency have been suggested to be the causes of increased RDWs in patients with CKD [40]. However, the factors that affect the increase in RDW are still unclear. The finding that HIF-PH inhibitors improve anemia without increasing RDW suggested that they may provide a different erythropoiesis environment than conventional ESAs. Therefore, these agents increase MCV and MCH without increasing iron stores while suppressing RDW. However, further studies are needed to determine the impact of HIF-PH inhibitors on adverse events and survival in patients with CKD who are receiving anemia treatment.

In a retrospective study including 26,626 patients with nondialysis-dependent CKD (>stage 3a) and Hb levels less than 11 g/dL [40], we demonstrated the importance of stable Hb management within the target Hb range to avoid the risks of premature death, cardiovascular events, dialysis introduction, and red blood cell transfusion. In addition, we also found that the cumulative incidence of discontinuation of anemia-related treatments with HIF-PH inhibitors (46.3%, 39.9 to 53.2%) was significantly lower than that of ESA, oral iron administration, or intravenous iron administration at 6 months after treatment initiation. Furthermore, there was no significant difference in the mean Hb level between patients treated with the ESA HIF-PH inhibitor at 3, 6, 9, or 12 months of follow-up. Although the patient population of Hb levels under 10 g/dL was decreased in patients treated with both ESA and HIF-PH inhibitors, at 12 months, 30.1% of patients were treated with ESA, and 15.6% were treated with HIF-PH inhibitors. The difference in the effect on patients with severe anemia that was confirmed in this study between ESA and HIF-PH may be caused by the route of administration (injection vs. oral), which is possibly due to the fact that the difference in the erythropoiesis environment between HIF-PH inhibitors and ESAs also affects patients with severe anemia.

## 6. Conclusions

Treatment of renal anemia has dramatically progressed due to the clinical use of ESAs. The results of many clinical studies continue to clarify the target Hb levels for patients with CKD. Recent studies have demonstrated that anemia associated with CKD is caused not only by a relative deficiency of EPO or iron deficiency but also by a variety of other factors, such as defective iron utilization for erythropoiesis. Furthermore, the issues of treating anemia with ESAs and iron supplements are being clarified. HIF-PH inhibitors, which have emerged as a new class of therapeutics for renal anemia, improve anemia through a mechanism that is different from that of conventional anemia drugs, such as the induction of endogenous EPO and improvement of defective iron utilization. These agents are expected to solve the problems of conventional anemia treatment; however, there are still some issues that need to be clarified, such as their safety in long-term use (Table 1). Now that a variety of drugs are available for renal anemia and much evidence supporting anemia treatments has accumulated, the target Hb level in patients with CKD could be maintained with the well-balanced use of ESAs, HIF-PH inhibitors, and iron preparations.

## Figures and Tables

**Table 1 ijms-25-07322-t001:** Comparison of Potential benefit and issues between ESAs and HIF-PH Inhibitors.

	Potential Benefits	Issues
**ESAs**	Abundant treatment experience Established long-term safety	Exogenous EPO agents Use for ESA hyporesponsiveness Unstable blood EPO levels
	Injection agents Accurate treatment effect evaluation Avoid poly pharmacy	Injection agents Increase hospital visits Pain at administration (peritoneal dialysis and not on dialysis patients)
**HIF-PH** **inhibitors**	Novel agents Maintain physiological EPO level Normalized erythrocyte indices (MCV↑, MCH↑RDW→) Improvement of iron absorption and iron utilization for erythropoiesis Avoid of iron overload	Novel agents Lack of evidence for long-term safety (malignancies, pulmonary arterial hypertensions, cyst growth in polycystic kidney disease, MACEs, thyroid dysfunction, etc.)
	Oral agent Reduce hospital visits No pain at administration (peritoneal dialysis and not on dialysis patients)	Oral agent Concern about decreased adherence Risk of polypharmacy Gastrointestinal symptoms

## Data Availability

Not applicable.

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
