# Peer review of "Treatment of Anemia Associated with Chronic Kidney Disease: Plea for Considering Physiological Erythropoiesis"

_ijms, 2024, doi:10.3390/ijms25137322_

Round 1

Reviewer 1 Report

Comments and Suggestions for Authors

Comments on “Treatment of anemia associated with chronic kidney disease 2 considering physiological erythropoiesis”

The well-known author discussed the ESA, Iron and HIF-PH in CKD

There are some commets:

1.      P2 L63, “in the fibroblasts of the renal tubular cortex”:

Fibroblasts of the interstitium in renal cortex?

2.      P8 L352, “Table 3”:

We did not find a table 2 or 3.

3.      Duplication of reference 31 and 32.

4.      In paragraph “Issues of anemia treatment with HIF-PH inhibitors”, the author did not mention the risk of MACE in PRO2TECT

The author should balance the report on the main trials. There may be difference between dialysis and non-dialysis.

In conclusion, the author summarized the risk of ESA and iron treatment and the advantage and disadvantage of HIF-PH inhibitor.

Author Response

Thank you very much for reviewing our manuscript and for providing us with valuable comments for its improvement. Your comments have been helpful in allowing us to revise our manuscript. We appreciate the time you have taken to suggest improvements for the manuscript.

  1. We have corrected according to Reviewer 1. From in the fibroblasts of the renal tubular cortex to fibroblasts of the interstitium in renal cortex.
  2. We have corrected table 1
  3. We have removed reference 32
  4. As Reviewer 1 recommended we have added the following sentence.

On the other hand, the PRO2TECT Clinical Trials (28), which conducted on the not on dialysis CKD patients, failed to demonstrate the non-inferiority of Vadadustat to darbepoetin in terms of risk of MACE. Furthermore, in a sub-analysis of this study, demonstrated non-inferior in the risk of MACE or expanded MACE of Vadadustat compared with darbepoetin studies conducted in the United States, but Vadadustat's non-inferiority was not demonstrated outside the United States.

  1. Chertow GM, Pergola PE, Farag YMK, Agarwal R, Arnold S, Bako G, Block GA, Burke S, Castillo FP, Jardine AG, Khawaja Z, Koury MJ, Lewis EF, Lin T, Luo W, Maroni BJ, Matsushita K, McCullough PA, Parfrey PS, Roy-Chaudhury P, Sarnak MJ, Sharma A, Spinowitz B, Tseng C, Tumlin J, Vargo DL, Walters KA, Winkelmayer WC, Wittes J, Eckardt KU; PRO2TECT Study Group. Vadadustat in Patients with Anemia and Non-Dialysis-Dependent CKD. N Engl J Med. 2021. 29;384(17):1589-1600.

Reviewer 2 Report

Comments and Suggestions for Authors

Kuranago submits a review entitled "Treatment of anemia associated with chronic kidney disease considering physiological erythropoiesis".

The subject is timely and interesting. Renal anemia is a field a bit neglegted, yet it is at the origin of many cardiovascular diseases in kidney patients. So the topic of erythropoiesis-stimulating agents (ESAs) or iron preparations in the treatment of anemia associated with chronic kidney disease (CKD) is stimulating.

In my opinion, the cellular and molecular causes of CKD-associated anmia are not developed enough in the paper, especialy as IJMS is a generalist journal, and not a medical one.  I would advise the author to write a chapter on the roles of red blood apoptosis and uremic toxins in the decrease of erythropoiesis in CKD patients (PMID: 32899941 and PMID: 36596353).

Can the authors discuss more how the target Hb level in patients with CKD should be maintained with the well-balanced use of ESAs, as an excess can be dangerous.

Also, one or two recapitulative figures would be useful as iconography of the article is poor.

Author Response

Reviewer 2

Thank you very much for reviewing our manuscript and for providing us with valuable comments for its improvement. Your comments have been helpful in allowing us to revise our manuscript. We appreciate the time you have taken to suggest improvements for the manuscript.

1.As Reviewer 2 suggested we have added following sentence in the Issues of hyporesponsiveness to ESAs area.Moreover, it has been reported that accumulation of uremic toxins such as Indoxyl sulfate in blood impairs erythropoietin synthesis, compromising the growth and differentiation of red blood cells in the bone marrow, leading to a subsequent impairment of erythropoiesis. Thus, the physiopathology of anemia in CKD focusing on a potential mechanism of EPO on iron regulation in uremic condition also might be important (5). Hamza E, Metzinger L, Metzinger-Le Meuth V. Uremic Toxins Affect Erythropoiesis during the Course of Chronic Kidney Disease: A Review. Cells 2020 6;9(9):2039. doi: 10.2020

2. As Reviewer 2 suggested the it is very difficult problem the treatment which kept target Hb level in patients with CKD with the well-balanced use of ESAs. Thus, we described in the conclusion area, to maintain target Hb in CKD patients, avoiding excessive use of ESAs, iron supplements, and HIF-PH inhibitors is very important.

3. As the reviewer pointed out, a single figure is poor. However, we currently have no ideas for new figures and tables, and I have little time to resubmit my paper, so we would appreciate your consideration.

Reviewer 3 Report

Comments and Suggestions for Authors

The manuscript is a review of the treatment of anemia in CKD, It provides valuable insights into both traditional and novel therapies. However, some improvements in critical analysis, clarity, and presentation are necessary to make it more useful as a resource for clinicians and researchers in the field.

Comments:

1.  While the abstract provides a summary, it lacks sufficient detail about the main findings and conclusions. A more detailed abstract would help readers quickly grasp the key points of the review

2. Discussing potential biases, limitations, and the quality of the evidence would strengthen the review’s credibility.

3.  The conclusions do not offer specific recommendations for clinical practice making it more useful for practitioners looking for guidance on implementing the findings in a clinical setting.

4.  While the manuscript highlights the need for long-term safety data on HIF-PH inhibitors, it does not delve deeply into existing long-term studies or potential areas for future research. This leaves a gap in the discussion of the long-term implications of newer therapies.

Author Response

Reviewer 3

Thank you very much for reviewing our manuscript and for providing us with valuable comments for its improvement. Your comments have been helpful in allowing us to revise our manuscript. We appreciate the time you have taken to suggest improvements for the manuscript.

  1. Our manuscript has been reviewed by American Journal expert.
  2. As Reviewer 3 recommended we have changed the title of this manuscript “Treatment of anemia associated with chronic kidney disease : plea for considering physiological erythropoiesis.
  3. As Reviewer 3 recommended we have added following sentence. PIVOTAL study also demonstrated that patients who enrolled in the proactive-high dose arm of this non-inferiority trial (per-protocol analysis) received an average monthly dose of 264 mg, whereas those in the low-dose reactive arm received 145 mg monthly, thus confirming prospectively the lack of toxicity in hemodialysis patients of intravenous iron dosage below 300 mg/month, which had been shown by DOPPS to be associated with mortality in this setting.
  1. As Reviewer 3 recommended we have changed following sentence. There is a high incidence of iron overload by liver magnetic resonance imaging among maintenance hemodialysis patients who are treated with intravenous iron preparations (19)”

Reviewer 4 Report

Comments and Suggestions for Authors

This interesting review performed by Prof Takahiro Kuragano depicts the actual limits of classical treatment of CKD anemia especially concerning patients with ESKD on dialysis, based on ESAs and iron therapy (mainly parenteral worldwide apart from Japan where oral iron is used as first line and parenteral iron as second line therapy) and make a wise plea for a more physiological approach of therapeutic modulation erythropoiesis especially with the new class of HIF-PH inhibitors.

Few comments to improve this high quality manuscript :

General comments

-       The English language should be reviewed by a native English or American translator

-       Since the iron therapy in Japan is far more conservative and very prudent with lower advised levels of ferritin and TSAT and lower usage of iron products, the author should contextualize for the readers where all the studies developed in deep have been performed (Japan, US, Europe, Asia, World (as DOPPS))

Specific comments

-       Title: given the philosophy of this text , I suggest,  the title to become : “Treatment of anemia associated with chronic kidney disease  : plea for considering physiological erythropoiesis”

-         Lines 121-125, the author wrote about the PIVOTAL study Furthermore, a multicenter randomized controlled trial of 2,141 hemodialysis patients demonstrated that, compared with those in a low-dose intravenous iron supplementation group (0-400 mg/month), those in a high-dose intravenous iron supplementation group (400 mg/month) exhibited a lower risk of death and nonfatal myocardial infarction (10)”. The PIVOTAL trial also confirmed that maintenance iron therapy was clearly superior to iron loading strategy. Mos importantly, patients of PIVOTAL study enrolled in the proactive-high dose arm of this non-inferiority trial (per-protocol analysis) received an average monthly dose of 264 mg, whereas those in the low-dose reactive arm received 145 mg monthly, thus confirming prospectively the lack of toxicity in patients on dialysis of IV iron dosage below 300 mg/month, which had been shown by DOPPS to be associated with mortality in this setting.

-       Paragraph Iron overload and high-dose iron administration- lines 207-208 please precise that these findings are made at liver MRI “ There is a high incidence of iron overload by liver magnetic resonance imaging among maintenance hemodialysis patients who are treated with intravenous iron preparations (19)”. 

Comments on the Quality of English Language

  The English language should be reviewed by a native English or American translator

Author Response

Reviewer 1
Thank you very much for reviewing our manuscript and for providing us with valuable comments for its improvement. Your comments have been helpful in allowing us to revise our manuscript. We appreciate the time you have taken to suggest improvements for the manuscript.
1. We have corrected according to Reviewer 1. From in the fibroblasts of the renal tubular cortex to fibroblasts of the interstitium in renal cortex.
2. We have corrected table 1
3. We have removed reference 32
4. As Reviewer 1 recommended we have added the following sentence.
On the other hand, the PRO2TECT Clinical Trials (28), which conducted on the not on dialysis CKD patients, failed to demonstrate the non-inferiority of Vadadustat to darbepoetin in terms of risk of MACE. Furthermore, in a sub-analysis of this study, demonstrated non-inferior in the risk of MACE or expanded MACE of Vadadustat compared with darbepoetin studies conducted in the United States, but Vadadustat's non-inferiority was not demonstrated outside the United States.
28. Chertow GM, Pergola PE, Farag YMK, Agarwal R, Arnold S, Bako G, Block GA, Burke S, Castillo FP, Jardine AG, Khawaja Z, Koury MJ, Lewis EF, Lin T, Luo W, Maroni BJ, Matsushita K, McCullough PA, Parfrey PS, Roy-Chaudhury P, Sarnak MJ, Sharma A, Spinowitz B, Tseng C, Tumlin J, Vargo DL, Walters KA, Winkelmayer WC, Wittes J, Eckardt KU; PRO2TECT Study Group. Vadadustat in Patients with Anemia and Non-Dialysis-Dependent CKD. N Engl J Med. 2021. 29;384(17):1589-1600.

Reviewer 2
Thank you very much for reviewing our manuscript and for providing us with valuable comments for its improvement. Your comments have been helpful in allowing us to revise our manuscript. We appreciate the time you have taken to suggest improvements for the manuscript.
1.As Reviewer 2 suggested we have added following sentence in the Issues of hyporesponsiveness to ESAs area.
Moreover, it has been reported that accumulation of uremic toxins such as Indoxyl sulfate in blood impairs erythropoietin synthesis, compromising the growth and differentiation of red blood cells in the bone marrow, leading to a subsequent impairment of erythropoiesis. Thus, the physiopathology of anemia in CKD focusing on a potential mechanism of EPO on iron regulation in uremic condition also might be important (5).  
5. Hamza E, Metzinger L, Metzinger-Le Meuth V. Uremic Toxins Affect Erythropoiesis during the Course of Chronic Kidney Disease: A Review. Cells 2020 6;9(9):2039. doi: 10.2020.
2. As Reviewer 2 suggested the it is very difficult problem the treatment which kept target Hb level in patients with CKD with the well-balanced use of ESAs.
Thus, we described in the conclusion area, to maintain target Hb in CKD patients, avoiding excessive use of ESAs, iron supplements, and HIF-PH inhibitors is very important. 
3. As the reviewer pointed out, a single figure is poor. However, we currently have no ideas for new figures and tables, and I have little time to resubmit my paper, so we would appreciate your consideration.

Reviewer 3
Thank you very much for reviewing our manuscript and for providing us with valuable comments for its improvement. Your comments have been helpful in allowing us to revise our manuscript. We appreciate the time you have taken to suggest improvements for the manuscript.
1. Our manuscript has been reviewed by American Journal expert.
2. As Reviewer 3 recommended we have changed the title of this manuscript “Treatment of anemia associated with chronic kidney disease  : plea for considering physiological erythropoiesis.
3. As Reviewer 3 recommended we have added following sentence.
PIVOTAL study also demonstrated that patients who enrolled in the proactive-high dose arm of this non-inferiority trial (per-protocol analysis) received an average monthly dose of 264 mg, whereas those in the low-dose reactive arm received 145 mg monthly, thus confirming prospectively the lack of toxicity in hemodialysis patients of intravenous iron dosage below 300 mg/month, which had been shown by DOPPS to be associated with mortality in this setting.
4. As Reviewer 3 recommended we have changed following sentence.
There is a high incidence of iron overload by liver magnetic resonance imaging among maintenance hemodialysis patients who are treated with intravenous iron preparations (19)”